# The Influence of Wŏnhyo's Understanding of "Shenjie" 神解 on the Chinese Commentaries on the *Awakening of Faith in Mahāyāna* †

**Jiyun Kim** 

Buddhist Culture Research Institute, Dongguk University, Seoul 04626, Republic of Korea;
dodododo8209@gmail.com

† This article is an edited transcript of a paper I presented at XIXth Congress of the International Association of Buddhist Studies (August 15–19, 2022).

**Abstract:** This study aims to reveal the influence of Wŏnhyo's *Kisillon so* (*Wŏnhyoso*) on Chinese commentaries on the *Awakening of Faith in Mahāyāna* (*AFM*), which is regarded as important in East Asian Buddhism. Previous studies focused only on the influence on Fazang's *Qixinlun shu* (*Fazangshu*), but it should be noted that the *Wŏnhyoso* also had an effect on the understanding of the *AFM* in China. First, by comparing the usage of "shenjie" in the *Fazangshu* and in the *Wŏnhyoso*, one can identify *Wŏnhyoso*'s unique interpretation. The *Wŏnhyoso* defines it as "mystical understanding as the nature of One Mind in the aspect of thusness and the nature of the mind of original enlightenment in the aspect of arising and ceasing", whereas the *Fazangshu* defines "shenjie" as "an excellent comprehension". Next, examining the usage of "shenjie" of the *Wŏnhyoso* in the later commentaries on the *AFM* after Fazang, such as the *Shilun*, the *Zanxuanshu*, the *Puguanji*, the *Zongmishu*, the *Bixueji*, the *Shulue*, and the *Huiyue*, has confirmed the influence of the *Wŏnhyoso* in Chinese Buddhism. In addition, the relationships between Chinese commentaries on the *AFM* were also clarified based on the commentaries' acceptance of "shenjie".

**Keywords:** shenjie 神解; *Wŏnhyoso*; *Fazangshu*; *Awakening of Faith in Mahāyāna*; Chinese commentaries on the *AFM*; relationship

## 1. Introduction

The *Awakening of Faith in Mahāyāna* 大乘起信論 (*Dasheng qixin lun*; hereinafter *AFM*) is regarded as a key text for understanding East Asian Buddhism. Many commentaries on the *AFM* were published in China, Korea, and Japan. In China, the *Dasheng qixinlun shu* 大乘起信論疏 of Kyo'u Library 杏雨書屋, the oldest commentary in existence, and the *Dasheng qixinlun yishu* 大乘起信論義疏 of Tanyan 曇延 (516–588) existed from the late fifth century. However, after Fazang's 法藏 (643–712) *Qixinlun shu* 起信論疏 (hereinafter *Fazangshu*) appeared, it received scholarly attention, and the awareness of the *AFM* also increased.[1] Since many scholars, such as Tankuang 曇曠 (700–788), Zongmi 宗密 (780–841), and Zixuan 子璿 (965–1038), annotated the *AFM* using the *Fazangshu* as their primary text, the *Fazangshu* became central to the study of the *AFM*.

However, that many sentences in the *Fazangshu* were referenced from the Silla monk Wŏnhyo's 元曉 (617–686) *Kisillon so* 起信論疏 (the commentary on the *AFM*, hereinafter *Wŏnhyoso*), which has already been revealed through previous research. Although these two documents are very similar in many parts, Fazang changed words or described sentences differently from the *Wŏnhyoso* in parts where he disagreed with Wŏnhyo. The interpretation of "shenjie" 神解 is one of the differences between the *Wŏnhyoso* and the *Fazangshu*.

"Shenjie" means spiritual or mystical understanding based on the meanings of "shen" 神, which is defined as supernatural and numinous, and "jie" 解, which refers to awakening or understanding.[2] Fazang's interpretation of "shenjie" is quite distinct from Wǒnhyo's. For this reason, Fazang used this word with a different meaning than the *Wǒnhyoso* in his narrative at the beginning of the *Fazangshu* and replaced the word "shenjie" with another word or did not quote sentences containing it when citing the *Wǒnhyoso*. However, the usage of "shenjie" in the *Fazangshu* is also distinguished from the commentaries on the *AFM* that have been strongly influenced by the *Fazangshu* such as the *Shi moheyan lun* 釋摩訶衍論 (hereinafter *Shilun*), T'aehyǒn's *Taesung kisillon naeuiyakt'amgi* 大乘起信論內義略探記, Zongmi's *Qixinlun shu* 起信論疏 (hereinafter *Zongmishu*), Zixuan's *Qixinlun shu bixueji* 起信論疏筆削記 (hereinafter *Bixueji*), Deqing's 德清 (1546–1623) *Dasheng qixinlun shulue* 大乘起信論疏略 (hereinafter *Shulue*), and Xufa's 續法 (1641–1728) *Qixinlun shuji huiyue* 起信論疏記會閱 (hereinafter *Huiyue*). They reflect the sentences and understanding of Wǒnhyo of "shenjie", although Fazang intentionally excluded them. This tendency verifies the influence of Wǒnhyo on the later Chinese commentators of the *AFM*. That is the reason this study focuses on "shenjie".

There are few studies on "shenjie" in spite of its importance. Ko (2008) examined the meaning of "shenjie" in all Wǒnhyo's writings and deduced that using "shenjie" broadened Wǒnhyo's definition of One Mind from the eighth consciousness *ālaya-vijñāna* to the ninth consciousness *amala-vijñāna*. Kim (2015, 2018) used "shenjie" as a basis for clarifying the relation between Zongmi and Wǒnhyo and the difference between Zongmi and Fazang by comparing the *Wǒnhyoso*, the *Fazangshu*, and the *Zongmishu*. Since this study focuses on Wǒnhyo's influence on the study of the *AFM* in China, I examine only the *Wǒnhyoso* among Wǒnhyo's writings and expand the research object to the commentaries on the *AFM* after Zongmi and the *Shilun* and its commentaries that have never been dealt with before.

First, I compare the usage of "shenjie" in the *Wǒnhyoso* and the *Fazangshu* to clarify the differences in understanding of "shenjie" of Wǒnhyo and Fazang. This includes revealing one of the unique characteristics of the *Wǒnhyoso*. Next, I categorize two groups of commentaries on the *AFM* as the *Shilun* and the *Zongmishu* and examine how Wǒnhyo's view of "shenjie" had influence on the commentaries on the *AFM* written after Fazang. Accordingly, the usage of "shenjie" elucidates the flow of thought from the *Wǒnhyoso* to the *Shilun* and the commentaries on the *Shilun*, the *Zongmishu*, the *Bixueji*, the *Shulue*, and the *Huiyue*. As a result, this study shows the genealogy of the commentaries on the *AFM* from the Tang to the Qing dynasties by defining their relations and will expand the area of research on the *AFM*.

## 2. The Different Usage of "Shenjie" between Wǒnhyo and Fazang

### 2.1. Fazang's View of "Shenjie"

Fazang follows the *Wǒnhyoso* in many parts but presents his own alternative interpretation when his opinion differs from Wǒnhyo, e.g., when he analyzed the concept of manas, shenjie, and so on. The different views on "shenjie" between Wǒnhyo and Fazang not only show the descriptive characteristics of the *Wǒnhyoso*, but also become the key to understanding the relations with the later commentaries on the *AFM*.

The *Fazangshu* used the word "shenjie" only once: "Two eminent treatise masters were contemporaries, one is Jiexian 戒賢 (*Śīlabhadra*), and the other is Zhiguang 智光 (*Jñānaprabha*). Their excellent comprehension transcended ordinary persons".[3] "Shenjie" describes the outstanding ability of Jiexian and Zhiguang. This usage of "shenjie" in the *Fazangshu* is the same as in the type of biographies, such as the *Xu gaoseng zhuan* 續高僧傳 (*Continued Biographies of Eminent Monks*) and the *Haedong kosǔngjǒn* 海東高僧傳 (*Lives of Eminent Korean Monks*), but is different from the *Wǒnhyoso*. So, how does Wǒnhyo interpret "shenjie"?

*2.2. The Meaning of "Shenjie" in the Wŏnhyoso*

The word "shenjie" is used six times in the *Wŏnhyoso*: ① T44.207a1 (once), ② T44.216c2 7 (once), ③ T44.208b18 (once), ④ T44.211b13 (once), ⑤ T44.208c9-10 (twice).

First, the *Wŏnhyoso* ① is the explanation of the *AFM*'s sentence, "The revelation of correct meaning is that there are two aspects relying on One Mind 一心 (yixin). What are the two? One is the aspect of mind of the thusness, and the other is the aspect of mind of arising and ceasing",[4] by using the question-and-answer format to explain the relationship between One Mind and two aspects, which are the thusness 眞如 (zhenru) and arising and ceasing 生滅 (shengmie).

> *Wŏnhyoso* ①
>
> Two aspects, [the mind in its aspect of "thusness" and "arising and ceasing"], are like this, how do they become One Mind? It is named "one" because the nature of all the defiled and pure dharmas is not two and there are no differences between the two aspects of true and false. [Then,] it is named "mind" since a place without discrimination between two is the true aspect of the middle way of all dharma and is not the same as space, and its nature understands mystically by itself.[5]

Wŏnhyo divides One Mind into "one" and "mind". The former means the nature of Dharma, and the latter expresses that nature understands mystically by itself. The phrase "nature understands mystically by itself" 性自神解 (xingzishenjie) has been circulated in East Asian Buddhism since Wŏnhyo first used it (Kim 2018, p. 50). It could be found more than 49 times in the *Zongjing lu* 宗鏡錄 (*Record of the Axiom Mirror*), the *Goryeoguk bojo seonsa susim gyeol* 高麗國普照禪師修心訣 (*Secrets on Cultivating the Mind*), etc.[6] From this, one aspect of the influence of the *Wŏnhyoso* throughout East Asian Buddhism is identifiable.

Second, Wŏnhyo defines "shenjie" as "awareness" 智 (zhi) when he accounts for the *AFM*'s statement, "Only the ignorance 癡 (chi) ceases, so the aspect of the mind also vanishes, but the awareness of mind 心智 (xinzhi) [of the original enlightenment] does not disappear".[7]

> *Wŏnhyoso* ②
>
> In the sentence, "The awareness of the mind [of the original enlightenment] does not disappear," "the awareness of the mind" indicates the nature of mystical understanding. It is the same as the above "The nature of awareness does not destroy," so it reveals the meaning that the unique characteristic 自相 (zixiang) does not become extinct.[8]

This description concerns the aspect of the defiled dharmas 染法相 (ranfaxiang) in the *AFM*. Wŏnhyo matches the *AFM*'s sentence, "The awareness of the mind [of the original enlightenment] does not disappear," with another of *AFM*'s sentences: "The nature of awareness is not destroyed". In this way, Wŏnhyo perceives "shenjie" as the awareness of the mind of original enlightenment. Therefore, "shenjie" is another expression of "awareness".

Wŏnhyo also clarifies that the unique characteristic does not become extinct with the *AFM*'s sentence, "The nature of awareness is not destroyed" (*Wŏnhyoso* ④). According to the *Wŏnhyoso* ⑤, the unique characteristic 自相 is the expression of the *Rulengqie jing* 入楞伽經 (*Laṅkâvatāra-sūtra*) and is identical to "the unique true characteristic" 自眞相 (zizhenxiang) of the *Lengqie abatuoluo baojing* 楞伽阿跋多羅寶經. The unique true characteristic is applied to both states of "neither arising nor ceasing" and "arising and ceasing". The awareness of the intrinsically enlightened mind of the *Wŏnhyoso* ② corresponds to the unique true characteristic of the "arising and ceasing" state.ṅ

Comparing the *Wǒnhyoso* ① and ②, Wǒnhyo expresses "shenjie" differently according to the aspect of thusness and the aspect of arising and ceasing. In the *Wǒnhyoso* ①, "nature" is the subject, and "shenjie" is an adverb and a verb meaning "understands mystically" because "shenjie" is represented in the mind of neither arising nor ceasing. In contrast, in the *Wǒnhyoso* ②, "shenjie" is an adjective and a noun meaning "mystical understanding" that modifies "nature" because "shenjie" is explained in the mind of arising and ceasing. In the *Wǒnhyoso*, "shenjie" is the awareness of One Mind and the nature of the mind of the original enlightenment that is not destroyed in the state of "neither arising nor ceasing" nor in the state of "arising and ceasing".

*2.3. Fazang's Perspective on Wǒnhyo's Understanding of "Shenjie"*

Fazang disagrees with Wǒnhyo's view of "shenjie" as the unchanging nature of One Mind because he intentionally excluded the word "shenjie" or the sentence including "shenjie". This could be confirmed by comparing the *Fazangshu* section corresponding to the *Wǒnhyoso* ③~⑤ below.

The *Wǒnhyoso* ③ accounts for the harmony between "neither arising nor ceasing" 不生不滅 (bushengbumie) and "arising and ceasing" 生滅. Wǒnhyo matches seawater 水 (shui) and movement 動 (dong) to "neither arising nor ceasing" and "arising and ceasing" by using the metaphor of waves in the latter part of the *AFM*. Then, he analyzes that the movement of seawater caused by the wind is the mark of the wind 風相 (fengxiang), and the moisture of the seawater that does not change even if the water moves is the mark of the water 水相 (shuixiang). In this context, Wǒnhyo connects the moisture to "shenjie" and shows the nature of "neither arising nor ceasing" in the arising and ceasing aspect.

The *Wǒnhyoso* ④ considers that the nature of awareness is equivalent to the nature of "shenjie" and compares the nature of awareness to the nature of the moisture of water. The metaphor of the waves in this part is exactly what was said in the *Wǒnhyoso* ③: "This is the same as the below sentence," and both similarly apply the moisture to "shenjie" [awareness].

Fazang consults the sentences of the *Wǒnhyoso* almost as it is in many parts but changes only "shenjie" to "zhen" 眞 (③) and "zhaocha" 照察 (④), as shown in Table 1. A similar tendency is shown in Table 2.

**Table 1.** Fazang's change to "shenjie".

| *Wǒnhyoso* | *Fazangshu* |
| --- | --- |
| 如下文言。如大海水因風波動。水相風相不相捨離。乃至廣說。此中水之動是風相。動之濕是水相。水舉體動。故水不離風相。無動非濕。故動不離水相。心亦如是。不生滅心舉體動。故心不離生滅相。生滅之相莫非神解。故生滅不離心相。(T44, 208b13-18) | 故下云。如大海水因風波動。水相風相不相捨離。乃至廣說。此中水之動是風相。動之濕是水相。以水舉體動故。水不離於風相。無動而非濕。故動不離於水相。心亦如是。不生滅心舉體動故。心不離生滅相。生滅之相莫非眞故。生滅不離於心相。(T44, 254c13-19) |

③  This is the same as the below sentence "As if the waves of a sea are moved by the wind, the mark of water and the mark of wind do not separate from each other" in the text below. In this sentence, the movement of seawater is the mark of the wind, and the moisture of the movement is the mark of seawater. The seawater does not lose the mark of the wind because all the seawater moves, and the moving wave does not separate from the mark of the seawater because there is no non-moisture in movement. The mind is like this; the mind does not lose the mark of arising and ceasing because the whole mind that does not arise and cease moves, and the mark of arising and ceasing does not separate from the mark of mind since there is no non-mystical understanding 非神解 [un-real 非眞] in the mark of arising and ceasing.

**Table 1.** *Cont.*

| *Wŏnhyoso* | *Fazangshu* |
|---|---|
| 合中言無明滅者。本無明滅。是合風滅也。相續即滅者。業識等滅。合動相滅也。智性不壞者。隨染本覺神解之性名爲智性。是合濕性不壞也。<br>(T44, 211b10-13) | 無明滅者。是根本無明滅。合風滅也。相續滅者。業識等滅。合動相滅。智性不壞者。　隨染本覺照察之性。是合濕性不壞。<br>(T44, 260b24-26) |

④ In application 合, "if the nescience 無明 (wuming) ceases" means the original nescience vanishes. It applies to the application of the phrase, "The wind stops". "The continuity ceases immediately" means that the karmic consciousness, etc., is ceasing. It applies to the phrase, "The nature of movement stops". "The nature of awareness is not destroyed" is the application of "The nature of moisture does not disappear", and the nature of awareness is the nature of <u>mystical understanding</u> 神解 [<u>clear observation</u> 照察 (zhaocha)].

**Table 2.** Fazang's exclusion of "shenjie".

| *Wŏnhyoso* | *Fazangshu* |
|---|---|
| 如是<u>轉識藏識眞相若異者。藏識非因若不異者。轉識滅藏識亦應滅。而自眞相實不滅。是故非自眞相識滅。但業相滅。今此論主正釋彼文。故言非一非異。</u><br>此中業識者。因無明力不覺心動。故名業識。又依動心轉成能見。故名轉識。此二皆在梨耶識位。如十卷經言。如來藏卽阿梨耶識。共七識生。名轉滅相。故知轉相在梨耶識。**自眞相者。十卷經云中眞名自相。本覺之心。不藉妄緣。性自神解名自眞相。是約不一義門說也。又隨無明風作生滅時。神解之性與本不異。故亦得名爲自眞相。是依不異義門說也。**於中委悉。如別記說也。<br><u>第三立名</u>。名爲阿梨耶識者。[9]<br>(T44. 208b29-c13) | 如是<u>轉識藏識眞相若異者。藏識非因。若不異者。轉識滅。藏識亦應滅。而自眞相實不滅。是故非自眞相識滅。但業相滅。</u>解云。此中眞相是如來藏轉識是七識。藏識是梨耶。<u>今此論主總括彼楞伽經上下文</u>意作此安立。<u>故云非一異</u>也。<br><br><br><br><br><br><u>第三立名</u>。名爲阿梨耶識。…… …… 。<br>(T44. 255b23-c1) |

　　In the *Wŏnhyoso* ⑤, "neither identical" and "nor different" are expressed as "not one" 不一 (buyi) and "not different" 不異 (buyi), and they explain the nature of the mind of the original enlightenment with the two aspects of "shenjie". The first is the state in which "arising and ceasing" does not occur. "Shenjie" describes the nature of the mind of the original enlightenment of "neither arising nor ceasing," which is different from "arising and ceasing". In this part, the sentence "the nature understands mysteriously by itself" 性自神解 is the same as the *Wŏnhyoso* ① that analyzes One Mind. The second is the state in which "arising and ceasing" is caused by nescience 無明. "Shenjie" represents the original nature of the mind of the original enlightenment that does not change in the arising and ceasing aspect 生滅相. It is compared to moisture, an aspect of water that is not destroyed even when the seawater moves, as in the *Wŏnhyoso* ③.

　　Like in Table 2, Fazang follows the interpretation of the *Wŏnhyoso* ⑤ but excludes the part that includes the word "shenjie". In addition, Fazang does not mention "the unique true characteristic," which is another expression of the mystical nature of the mind of the original enlightenment. Fazang did not use the word "shenjie" intentionally based on *Fazangshu*'s inclination to change "shenjie" to another word or to omit the part of the explanation about "shenjie".

　　The different views on "shenjie" between Wŏnhyo and Fazang not only proclaim the character of the *Wŏnhyoso*, but also are key to understanding the relationship with the later commentaries on the *AFM*.

### 3. The Usage of "Shenjie" in the Commentaries after Fazang

Since Fazang, it is not an exaggeration to say that several commentaries on the *AFM* understood their main text through the *Fazangshu*, so the *Fazangshu* significantly influenced the *AFM* study. There could have been an indirect effect through Wŏnhyo's interpretation quoted in the *Fazangshu* as the *Fazangshu* became popular. However, there is a direct effect of the *Wŏnhyoso* because the word "shenjie," which Fazang intentionally excluded, is found in the *Shilun*, the *Zongmishu*, the *Bixueji*, the *Shulue*, and the *Huiyue*.

Among them, the *Bixueji*, the *Shulue*, and the *Huiyue* were strongly influenced by the *Zongmishu*, which refers to the *Fazangshu* as central and references the *Wŏnhyoso* directly. Therefore, in this chapter, the effect of the *Wŏnhyoso* on the commentaries on the *AFM* after Fazang is examined in detail by classifying it into two groups: the *Shilun* and the *Zongmishu*.[10]

#### 3.1. Distinction between Consciousness and Mark

The *Shilun* is one of the commentaries on the *AFM*.[11] The reason why we should pay attention to the *Shilun* is that this treatise had an effect on the Buddhist study of the day by forming an academic trend, as several commentaries on the *Shilun* were made in China and Japan throughout the ages.

The *Shilun* used the "shenjie" only twice in the part that explains the five kinds of consciousness 五意 (wuyi) of the *AFM* as below.

> *Shilun*
>
> All defiled dharma has two meanings respectively. What are the two meanings? The first meaning is <u>mystical understanding</u>, and the second is dark and dull. In terms of the continuous arising from the original enlightenment, it sets up the meaning of <u>mystical understanding</u>. Then, in terms of the continuous arising from nescience, it sets up the meaning of dark and dull. Based on the first aspect, the name "consciousness" is given. Based on the second aspect, the name "mark" is given. You should know the truth about the difference between the two aspects as above. How do they have distinctive characteristics? Consciousness conforms to the original enlightenment because it means "understanding" and "enlightenment". On the other hand, the mark follows the nescience since it signifies "to betray the original enlightenment".[12]

The *Shilun* suggests two significations of the defiled dharmas, mystical understanding and dull, and distinguishes consciousness from the mark based on their meaning. The word "shenjie" is used two times as an expression to suggest that the original enlightenment is the essence of the defiled dharmas, which was created by the movement of the original enlightenment following the nescience.

However, the parts of the *AFM* that the *Wŏnhyoso* and the *Shilun* use "shenjie" to explain are different, and the *Shilun* does not use the *Wŏnhyoso*'s expressions, such as "the nature understands mystically by itself" 性自神解 or "the nature of mystical understanding" 神解之性 (shenjiezhixing). Moreover, the *Shilun* comes up with a new interpretation, which the *Wŏnhyoso* does not mention, that "shenjie" of the original enlightenment 本覺 (benjue) is contrasted with the imbecility of the nescience, and each corresponds to consciousness 識 (shi) and mark 相, respectively. However, it is possible to infer that the *Shilun* is affected by the *Wŏnhyoso* because the usage of "shenjie" in the *Shilun* shows that the essence 體 (ti) of defiled dharmas 染法 is the original enlightenment 本覺.

The influence of the *Wŏnhyoso* reflected in the *Shilun* continues in several commentaries on the *Shilun* such as *Shi moheyan lun zanxuanshu* 釋摩訶衍論贊玄疏 (hereinafter *Zanxuanshu*) and *Shi moheyan lun ji* 釋摩訶衍論記 (hereinafter *Puguanji*). First, the *Zanxuanshu* was written by Fawu 法悟 in the Liao Dynasty and is composed of two parts; one is a summary part that divides the *Shilun*'s content into ten, and another is a detailed exposition part that interprets each of the *Shilun*'s sentences. The word "shenjie" is found four times

in the *Zanxuanshu*; one is used in the former and three times are in the latter.[13] Among the four cases, the fourth could be found in the quotation below.

> *Zanxuanshu*
>
> If the three main causes and indirect causes of defilement and purity are connected to the three subtle consciousnesses, the original enlightenment is the cause of proximity and the nescience is the condition of remoteness. Therefore, the result of the mystical understanding which is similar to the enlightenment occurs. [If the three main causes and indirect causes of defilement and purity are] related to the three subtle marks, the nescience is the cause of proximity and the original enlightenment is the condition of remoteness. Thus, the dharma of darkness which is similar to the nescience arises.[14]

The *Zanxuanshu* is an explanation of the sentence of the *Shilun* "以何(至)由疎爲緣故".[15] Fawu, the author of the *Zanxuanshu* adds his own description of the cause 因 (yin) of proximity and the condition 緣 (yuan) of remoteness by specifying the condition that distinguishes consciousness 識 and mark 相. However, the basic concept of the *Zanxuanshu* reflects the *Shilun*'s view that "shenjie" is connected to the original enlightenment and corresponds to consciousness.

Second, the *Puguanji* was written by Puguan's 普觀 in the Southern Song Dynasty. The word "shenjie" is used five times, four of which accounts for the ālaya-vijñāna 阿梨耶識 (aliyeshi) of the *AFM*'s sentence "the arising and ceasing mind means there is the arising and ceasing mind because [the mind] relies on the tathāgata-garbha 如來藏 (rulaizang). "'Neither arising nor ceasing' 不生不滅 combines with the 'arising and ceasing' 生滅, so [both are] neither identical nor different. That is called 'ālaya-vijñāna'".[16]

> *Puguanji*
>
> The fifth is [the ālaya-vijñāna] of the mark of karma and the activity consciousness. The mark is dark and dull, and consciousness is the mystical understanding ... The sixth is [the ālaya-vijñāna] of the mark of the subjective perceiver and the forthcoming consciousness... The visibility 有見 (youjian) is named consciousness because it relates the mystical understanding. The invisibility 無見 (wujian) is named mark since it relates the dark and dull. The seventh is [the ālaya-vijñāna] of the mark of the objective world and the manifesting consciousness... In addition, it is named consciousness that they are different respectively because it relates the mystical understanding. Then, it is named mark that they vary from each other since it relates the dark and dull... The tenth is [the ālaya-vijñāna] of the initial enlightenment of defilement and purity... Question: Two original enlightenment, two initial enlightenment, and nature as thusness 性真如 (xingzhenru) are called consciousness, but why is not it the same as the space as unconditioned 虛空無爲 (xukongwuwei)? Answer: Consciousness means mystical understanding is lucidity, [but] space is dark and dull and the function of the nescience is obvious. Therefore, it does not call [consciousness].[17]

The *Puguanji* accounts for the ten kinds of ālaya-vijñāna of the *Shilun*. Among them, the above paragraph is an interpretation of the fifth ālaya-vijñāna of the mark of karma and the activity consciousness 業相業識阿梨耶識 (yexiangyeshialiyeshi), the sixth ālaya-vijñāna of the mark of the subjective perceiver and the forthcoming consciousness 轉相轉識阿梨耶識 (zhuanxiangzhuanshialiyeshi), the seventh ālaya-vijñāna of the mark of the objective world and the manifesting consciousness 現相現識阿梨耶識 (xianxiangxianshialiyeshi), and the tenth ālaya-vijñāna of the initial enlightenment of defilement and purity 染淨始覺阿梨耶識 (ranjingshijuealiyeshi).

In the fifth ālaya-vijñāna, the Puguan discriminates between consciousness and the mark following the *Shilun*'s definition. In the sixth and the seventh, consciousness and the mark are distinguished by using terms such as visibility 有見, nihilism 無見, distinction 別異 (bieyi), and characteristics differ 相異 (xiangyi), which are mentioned in the sutras

quoted in the *Shilun*. In the tenth, he explains the reason why only the space as unconditioned 虛空無爲 and does not say "consciousness" among the four unconditioned factors 四無爲 (siwuwei), which reflects the unique interpretation of the *Shilun* through a question-and-answer format.

The *Puguanji* embraces the point of view in these four parts that the *Shilun* is to match "shenjie" to consciousness. However, although the *Shilun* refers to "shenjie" with the original enlightenment and consciousness, the *Puguanji* mentions only "shenjie" and consciousness. From this, it could be inferred that the *Puguanji* is more concerned with the relationship between "shenjie" and consciousness than "shenjie" and the original enlightenment.

In summary, the understanding of the *Wŏnhyoso* that "shenjie" is the nature of the original enlightenment has a prominent part in the *Shilun*. In addition, consciousness is added to the relationship between "shenjie" and original enlightenment by the *Shilun*, and this connection is reflected in the *Zanxuanshu* and the *Puguanji*. In other words, Wŏnhyo's interpretation of "shenjie" was handed down to the exegetist of the *Shilun*.

On the other hand, the other commentaries on the *Shilun*, the *Shi Moheyan lun shu* 釋摩訶衍論疏 of Famin 法敏, and the *Shi Moheyan lun ji* 釋摩訶衍論記 of Shengfa 聖法, do not deal with "shenjie", and the *Shi moheyan lun tongxuanchao* 釋摩訶衍論通玄鈔 of Zhifu 志福 used only the word "shen" 神 instead of the word "shenjie" when he distinguished consciousness and the mark. From this, it could be seen that Shengfa and Zhifu did not accept "shenjie" even though it needs further research whether they were influenced by Fazang or not. Therefore, it could be confirmed that the influence of the *Wŏnhyoso* is continued through the *Shilun* to the *Zanxuanshu* and the *Puguanji* since the use of "shenjie" is determined according to the opinion of commentators of the *Shilun*.

### 3.2. The Transmission of the Wŏnhyo's Understanding of "Shenjie" through the Zongmishu

Among the commentaries on the *AFM* written after Fazang in China, the word "shenjie" appears for the first time in the *Zongmishu*, which commented on the *Fazangshu* as the main text by Zongmi. The word "shenjie" is used twice in the *Zongmishu*, and the sentences containing "shenjie" are quoted from the *Wŏnhyoso* as Table 3 (Kim 2015, p. 52). These are Zongmi's restoration of the word "shenjie" of the *Wŏnhyoso* that Fazang excluded intentionally.

**Table 3.** Use of "shenjie" in the *Zongmishu*[18].

| | *Wŏnhyoso* | *Fazangshu* | *Zongmishu* |
|---|---|---|---|
| ① | 謂染淨諸法其性無二 真妄二門不得有異 故名為一。此無二處 諸法中實 不同虛空 性自神解 故名為心。(T44, 206c28-207a1) | 然此二門 舉體通融 際限不分 體相莫二。難以名目 故曰一心有二門等也。(T44, 251c) | 然此二門舉體通融際限不分 體相莫二。此無二處 諸法中實 不同虛空 性自神解 故云一心。(L141, 94b) |
| ③ | 心亦如是。不生滅心舉體動。故心不離生滅相。生滅之相莫非**神解**。故生滅不離心相。(T44, 208b16-18) | 心亦如是。不生滅心舉體動故。心不離生滅相。生滅之相莫非眞故。生滅不離於心相。(T44, 254c17-19) | 心亦如是。不生滅心舉體動故。心不離生滅之相生滅之相莫非**神解**。故生滅不離於心相。(L141, 98a10-11) |

Although Chengguan 澄觀 mentions the word "shenjie" in the *Dafangguang fo huanyan jing suishu yanyi chao* 大方廣佛華嚴經隨疏演義鈔 (*Subcommentary and Explanation of the Meaning of the Huanyan jing*) before Zongmi, Chengguan cited a different sentence of the *Wŏnhyoso*, including "shenjie" from the *Zongmishu*.[19] Therefore, it could be confirmed again that Zongmi directly referred to the *Wŏnhyoso*.

The reason why the *Zongmishu* is important in the study of "shenjie" in the commentaries on the *AFM* is that the *Zongmishu* shows the direct influence of the *Wŏnhyoso*. Moreover, beginning with the *Zongmishu*, Wŏnhyo's understanding of "shenjie" is inherited as

commentaries on the *AMF* such as the *Bixueji*, the *Shulue*, and the *Huiyue* written after Zongmi.

First, in Zixuan's *Bixueji*, the word "shenjie" is used three times. Zixuan wrote down only the first few letters of the passage instead of quoting *Zonggmishu*'s sentence as it is and then described his interpretation of it. The *Bixueji*① below is the explanation of the *Zongmishu*'s sentence cited not from the *Fazangshu*, but the *Wǒnhyoso* ①, "[a place without discrimination between two] is not the same as space, and its nature understands mystically by itself" 不同虛空 性自神解 (butongxukong xingzishenjie).

> *Bixueji* ①
>
> Below "not the same" 不同 (butong) is about understanding the mind by grasping the mystical illumination, that is the essence of the space 虛空 has no two borders 邊 (bian) and is not the distinctive deluded mark. Although there was only darkness and no mystical illumination before, now true nature is omnipotent and numinous penetration, so it is enlightened and is not dark. Therefore, it is said "butong"... It is called "one" because the essence and the attributes are not two, and it is said "mind" since it is the true aspect of the middle way 中實 (zhongshi) and the <u>mystical understanding</u>.[20]

Even though the *Bixueji* ① did not mention the full sentence of the *Wǒnhyoso* and just represented it as the word "butong", it is assumed that the Zixuan agrees with the Wǒnhyo's view of "shenjie" based on describing the relationship between "shenjie" and One Mind in the *Bixueji* ①. If not, "shenjie" would have been deleted or replaced with another word since the *Bixueji* added or took away some sentences from the *Dashengqixinlunsui shuji* 大乘起信論隨疏 of Chuan'ao 傳奧 (d.u.), which is the commentary on the *Zongmishu*. On the contrary to this, there is no word "shenjie" in some commentaries on the *Bixueji* published in the Ming 明 Dynasty, such as the *Qixinlun zuanzhu* 起信論纂註 of Zhenjie 真界 and the *Qixinlun jieyao* 起信論捷要 of Zhengyuan 正遠, so it shows that both disagree with Wǒnhyo's perspective on it.

Furthermore, through the *Bixueji* ② and the *Bixueji* ③, it is able to confirm the fact that the *Bixueji* inherits the point of view on "shenjie" from Wǒnhyo.

> *Bixueji* ②
>
> The part below "the mind also" is the third which is the application of the dharma. The <u>mystical understanding</u> is the psychomancy of penetrating discernment and the <u>lack of darkness of the original enlightenment</u>. The rest of the sentence could be understood.[21]

> *Bixueji* ③
>
> There are many ways to arrive at the true way of nirvana, the key point is śamatha 止 and vipaśyanā 觀 (guan). The śamatha is the first aspect to defeat defilements, and the vipaśyanā is the right way to break off delusion. The śamatha cultivates a good foundation of the mind and consciousness, and the vipaśyanā illuminates the marvelous skill of <u>mystical understanding</u>.[22]

The *Bixueji* ② corresponds to the *Zongmishu* [=the *Wǒnhyoso* ②] and explains "shenjie" as "the psychomancy of penetrating discernment and the lack of darkness of the original enlightenment". The *Bixueji* ③ accounts for *Śamatha* and *Vipaśyanā* Meditation of the *AFM* by quoting Zhiyi's 智顗 *Xiuxi zhiguan zuochan fayao* 修習止觀坐禪法要 (*Brief Clarification of the Essentials of Śamatha and Vipaśyanā Meditation for Beginners to Open their Blind Eyes*). The *Bixueji* ② and the *Bixueji* ③ are Zixuan's own definition, which is in neither the *Zongmishu* nor the *Wǒnhyoso*. From this, it could be verified that the Wǒnhyo's interpretation of "shenjie" developed further as it passed from the *Zongmishu* to the *Bixueji*.

Second is the *Shulue*, and the author Deqing states that he has edited the *Fazangshu* and made it briefly.[23] However, the word "shenjie", which is not used by Fazang, is found twice in the *Shulue*.

As seen in Tables 4 and 5, the *Shulue*'s phrases "故一心云 (guyixinyun)" (Table 4), "下文云 (xiawenyun)" (Table 5), and "如是不離 名爲和合 (rushibuli mingweihehe)" (Table 5) are the same as the *Zongmishu* but are not mentioned in the *Wŏnhyoso* and the *Fazangshu*. Based on this fact, "shenjie" in the *Shulue* also comes from the *Zongmishu*, which accepted Wŏnhyo's idea.

**Table 4.** Use of "shenjie" in the *Shulue* 1.

| *Wŏnhyoso* | *Fazangshu* | *Zongmishu* | *Shulue* |
|---|---|---|---|
| 謂染淨諸法其性無二 真妄二門不得有異 故名為一。<br>此無二處 諸法中實不同虛空 性自**神解**。故名為心。<br>(T44, 206c-207a) | 然此二門 舉體通融際限不分 體相莫二難以名目 故曰一心有二門等也。<br>(T44, 251c) | 然此二門 舉體通融際限不分 體相莫二。<br><br>此無二處 諸法中實不同虛空 性自**神解**故云一心。<br>(L141, 94b) | 然此二門 舉體通融 體相莫二。<br><br>此無二處 諸法中實 不同虛空 性自**神解**故云一心。<br>(X45, 448a) |

**Table 5.** Use of "shenjie" in the *Shulue* 2.

| *Wŏnhyoso* | *Fazangshu* | *Zongmishu* | *Shulue* |
|---|---|---|---|
| 如大海水因風波動。水相風相不相捨離。乃至廣說。此中水之動是風相。動之濕是水相。水舉體動。故水不離風相。無動非濕。故動不離水相。<br><br>心亦如是。不生滅心舉體動。故心不離生滅相。生滅之相莫非**神解**。故生滅不離心相。<br>(T44, 208b13-19) | 如大海水因風波動。水相風相不相捨離。乃至廣說。此中水之動是風相。動之濕是水相。以水舉體動。故水不離於風相。無動而非濕。故動不離於水相。<br><br>心亦如是。不生滅心舉體動。故心不離生滅相。生滅之相莫非**眞**。故生滅不離於心相。<br>(T44, 254c13-19) | 下文云<br>如大海水因風波動。水相風相不相捨離。乃至廣說。此中水之動是風相。動之濕是水相。以水舉體動故。水不離於風相。無動而非濕。故動不離於水相。<br>心亦如是。不生滅心舉體動。故心不離生滅之相。生滅之相莫非**神解**。故生滅不離於心相。<br>如是不離 名為和合。<br>(L141, 98a8-11) | 下文云<br>如大海水因風波動。水相風相不相捨離。謂真心舉體成。<br><br><br><br>生滅之相。生滅之相莫非**神解**。不離真心<br>如是不離 名為和合。<br>(X45, 450c5-7) |

There are two possibilities. One is that the *Shulue* summarizes the *Fazangshu* by referring to the *Zongmishu* (Kim 2015, p. 47). Another is that Deqing's *Shulue* abridges the *Zongmishu*, which has been mistakenly known as the *Fazangshu* (Kim 2021, p. 22). During the Ming Dynasty, the era of Deqing, the same cases are discovered in some texts such as Zhenjian's 真鑑 *Lengyan jing zhengmaishu xuanshi* 楞嚴經正脉疏懸示 (*Commentary on the Śūraṃgama-sūtra*) and Zhengmi's 正謐 *Shi buermen zhiyao chao xiangjie* 十不二門指要鈔詳解 (*Explanation of Ten Aspects to Nonduality in the Tiantai School*).[24] In either case, it demonstrates that the *Shulue* was influenced by the *Zongmishu*, and Zongmi's understanding of "shenjie" of the *Wŏnhyoso* continues until the *Shulue*.

Third, Xufa s *Huiyue* is the compilation of the *Fazangshu* and the *Bixueji*, and they are abbreviated as "Shu" and "Ji". In the *Huiyue*, the word "shenjie" is used five times. Among them, three times are in the "Ji" part, which is the same as the *Bixueji* exactly (Table 6), and the last are found twice in the "Shu" part, which corresponds not to the usage of the *Fazangshu*, but the *Wŏnhyoso* (Table 7).

**Table 6.** Use of "shenjie" in the "Ji" part of the *Huiyue*.

|  | *Bixueji* | *Huiyue* |
|---|---|---|
| ① | 故祖師云。空寂體上自有本智。能知知之一字。眾妙之門。大抵意云。於一切染淨融通法中。有真實之體。了然鑒覺。目之為心。斯則體相不二故。云一中實**神解**故云心。(T44. 330a27-b2) | 故祖師云。空寂體上。自有本智能知。知之一字。眾妙之門。大抵意云。於一切染淨融通法中。有真實之體。了然鑒覺。目之為心。斯則體相不二。故云一中實**神解**。故云心。(X45, 593b5-8) |
| ② | "心亦"下三法合。**神解**者。本覺不昧。鑒照靈通也。(T44. 337b9-10) | 心亦下。三。法合。**神解**者。謂本覺不昧。鑒照靈通也。(X45, 604c24-605a1) |
| ③ | 故彼云。涅槃真法入乃多塗。論其急要不過止觀。止乃伏結之初門。觀乃斷惑之正要。止乃養心識之善資。觀則照**神解**之妙術等。若人成就定慧二法。斯乃自利利人。法無不備也。今之學流焉可偏習。(T44, 406a12-17) | 故彼文云。涅槃真法。入乃多途。論其急要。不過止觀。止乃伏結之初門。觀乃斷惑之正要。止乃養心識之善資。觀則照**神解**之妙術等。若人成就定慧二法。斯乃自利利人。法無不備也。今之學流。焉可偏習。(X45, 725b16-20) |

**Table 7.** Use of "shenjie" in the "shu" part of the *Huiyue*.

|  | *Wŏnhyoso* | *Fazangshu* | *Zonghmishu* | *Huiyue* |
|---|---|---|---|---|
| ① | 謂染淨諸法其性無二 真妄二門不得有異 故名為一。此無二處 諸法中實不同虛空 性自神解故云一心。(T44,206c28-207a1) | 然此二門 舉體通融 際限不分 體相莫二。難以名目 故曰一心 有二門等也。(T44, 251c) | 然此二門 舉體通融 際限不分 體相莫二。此無二處 諸法中實不同虛空 性自神解故云一心。(L141, 94b) | 【疏】然此二門 舉體通融 際限不分 體相莫二。此無二處 諸法中實不同虛空 性自神解故云一心。(X45, 592a21-22) |
| ③ | 心亦如是。不生滅心舉體動故。心不離生滅相。生滅之相莫非神解故。生滅不離心相。如是不相離。故名與和合。(T44, 208b16-19) | 心亦如是。不生滅心舉體動故。心不離生滅相。生滅之相莫非真故。生滅不離於心相。如是不離 名為和合。(T44, 254c17-20) | 心亦如是。不生滅心舉體動故。心不離生滅之相。生滅之相莫非神解故。生滅不離於心相。如是不離 名為和合。(L141, 98a10-11) | 心亦如是。不生滅心舉體動故。心不離於生滅之相。生滅之相莫非神解故。生滅不離於心相。如是不離。名為和合。(X45, 604c13-15) |

Like the *Shulue*, it is assumed that the *Huiyue* referred to the *Zongmishu*, which is considered the *Fazangshu*. The ground is that the sentences with the phrase "Fazang says", which Xufa cited in his other writings such as the *Guanzizai pusa ruyilun tuoluoni jing lueshu* 觀自在菩薩如意心陀羅尼經略疏 (*Abbreviated Commentary on the Dhāraṇī Spell of the Wish-Fulfilling Essence of the Bodhisattva of Spontaneous Contemplation*) and the *Ba daren jue jing shu* 八大人覺經疏 (*Commentary on the Sutra on the Eight Kinds of Attentiveness of Great Persons*), are also the sentences of the *Zongmishu* (Kim 2021, p. 81). It infers another possibility that "shenjie" of the *Zongmishu*, which is inherited from the *Wŏnhyoso*, was considered as *Fazangshu*'s idea as well because people mistook the *Zongmishu* for the *Fazangshu*.

## 4. Conclusions

This study aims to reveal the influence of the *Wŏnhyoso* on Chinese commentaries on the *AFM*. To summarize, two points must be considered.

The first point concerns the distinctive interpretation of "shenjie" of Wŏnhyo from Fazang about the *AFM*. Wŏnhyo comprehends "shenjie" as the nature of One Mind in the aspect of thusness and the nature of the mind of original enlightenment in the aspect of arising and ceasing. On the other hand, Fazang uses "shenjie" to refer to a person of high intelligence. In addition, after thoroughly reviewing the part where Wŏnhyo interprets the *AFM* with "shenjie", Fazang changes "shenjie" to another word, such as "real," or replaces Wŏnhyo's sentences with his statement without "shenjie".

The second is how the understanding of "shenjie" of the *Wŏnhyoso* was accepted in the commentaries on the *AFM* after Fazang. The *Shilun*, like the *Wŏnhyoso*, explains "shenjie" as the nature of original enlightenment that exists in the defiled dharma. Furthermore, the *Shilun* relates "shenjie" to consciousness and contrasts between "shenjie" and nescience according to his own interpretation. Then, the commentaries on the *Shilun*, the *Zanxuanshu*, and the *Puguanji* follow the understanding of the *Wŏnhyoso*. However, the *Zanxuanshu* adds the condition for consciousness that is the distance between the cause and the condition. Moreover, the *Puguanji* mentions only the connection between "shenjie" and consciousness without the relation to original enlightenment. The *Zongmishu* interprets the *Fazangshu* while quoting *Wŏnhyoso*'s mention of "shenjie" excluded by Fazang. The *Bixueji* interprets *Zongmishu*'s sentence and adds the expression modifying "shenjie", psychomancy of penetrating discernment, and the relationship between "shenjie" and *Vipaśyanā*. Examining the usage of "shenjie" in the *Shulue* and the *Huiyue* suggests that both may refer to the *Zongmishu*, which has been mistakenly understood as the *Fazangshu* by scholars in the Ming dynasty.

In conclusion, this study reveals that "shenjie" is one of the keywords showing the differentiation between Wŏnhyo's and Fazang's perspectives. Examining the usage of "shenjie" clarifies the direct (the *Shilun* and the *Zongmishu*) and indirect (the *Zanxuanshu*, the *Puguanji*, the *Bixueji*, the *Shulue*, and the *Huiyue*) effects of the *Wŏnhyoso* on later commentaries on the *AFM*. In addition, reviewing the commentaries on the *AFM* shows that the meaning of "shenjie" has expanded over time from the Tang to the Qing dynasties and elucidates the relationship between the commentaries, as shown in Figure 1.

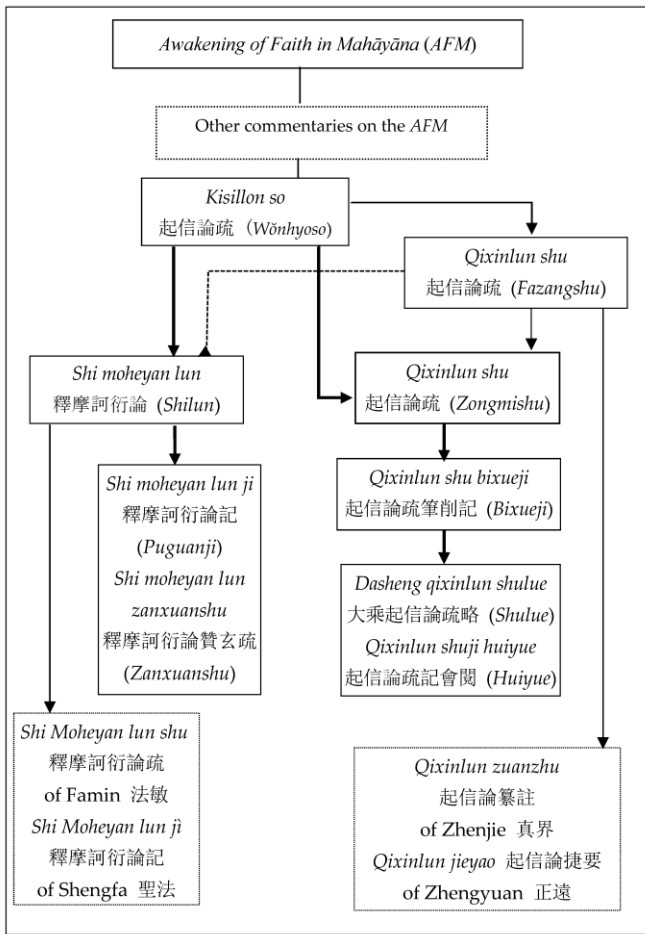

**Figure 1.** Genealogy of the commentaries on the *AFM* according to "shenjie".

**Funding:** This work was supported by the Ministry of Education of the Republic of Korea and the National Research Foundation of Korea (NRF-2021S1A6A3A01097807).

**Data Availability Statement:** Not applicable.

**Conflicts of Interest:** The author declares no conflict of interest.

## Abbreviations

| | |
|---|---|
| *AFM* | *Awakening of Faith in Mahāyāna* 大乘起信論 |
| *Bixueji* | *Qixinlun shu bixueji* 起信論疏筆削記 |
| *Fazangshu* | *Qixinlun shu* 起信論疏 of Fazang 法藏 |
| *Huiyue* | *Qixinlun shuji huiyue* 起信論疏記會閱 |
| L | *Qianlong dazing jing* 乾隆大藏經 |
| *Puguanji* | *Shi moheyan lun ji* 釋摩訶衍論記 |
| *Shilun* | *Shi moheyan lun* 釋摩訶衍論 |
| *Shulue* | *Dasheng qixinlun shulue* 大乘起信論疏略 |
| T | *Taishō shinshū daizōkyō* 大正新脩大藏經 |
| *Wŏnhyoso* | *Kisillon so* 起信論疏 of Wŏnhyo 元曉 |
| X | *Manji zokuzōkyō* 卍續藏經 |
| *Zanxuanshu* | *Shi moheyan lun zanxuanshu* 釋摩訶衍論贊玄疏 |
| *Zongmishu* | *Qixinlun shu* 起信論疏 of Zongmi 宗密 |

## Notes

1   Fazang's *Qixinlun shu* is written down as *Dasheng qixinlun yiji* in the *Taishō shinshū daizōkyō* 大正新脩大藏經. However, based on the result of the examination of the commentaries on the *Qixinlun shu* and the literature that quoted it, it is revealed that the original title is "Qixinlun shu". Therefore, in this paper, Fazang's commentary on the *AFM* is referred to as "Qixinlun shu". See (Kim 2018, 2021).

2   The word "shenjie" could be found in various works, such as the *Da banniepan jing jijie* 大般涅槃經集解 (*Compilation of Commentaries on the Nirvana Sutra*) and the *Weimo jing lue shou* 維摩經略疏 (*Abbreviated Commentary on the Vimalakīrti-nirdeśa-sūtra*). Since the scope is too wide, this study is limited to the commentaries on the *AFM*. In addition, the word "shen" 神 has many meaning in China. See (Kim 2006).

3   『大乘起信論義記』 (T44, 242b2-3), "同時有二大德論師。一曰戒賢。一曰智光。並神解超倫。" This sentence is mentioned in Fazang's other writings, such as the *Huayanjing tanxuan ji* 華嚴經探玄記 (*Record of the Search for the Profundities of the Huayan Sutra*, T35,111c12-14) and the *Shiermenlun zongzhi yiji* 十二門論宗致義記 (*Commentary on the Dvādaśanikāya-śāstra*, T42.213a7-8). In addition, this is quoted in later works after Fazang such as Zongmi's *Yuanjuejing dashu* 圓覺經大疏 (*Great Commentary on the Sutra of Perfect Enlightenment*, X9.327c14-15) and Purui's 普瑞 *Huayan xuan tanhui xuanji* 華嚴懸談會玄記 (*Commentary on the Flower Ornament Sutra*, X8.250c7-8).

4   『大乘起信論』 (T32, 576a4-6), "顯示正義者。依一心法。有二種門。云何為二。一者心真如門。二者心生滅門。" [The English translation of the *AFM* refers to (Hakeda 1967).]

5   『起信論疏』 (T44, 206c27-207a1), "二門如是。何為一心。謂染淨諸法其性無二。真妄二門不得有異。故名爲一。此無二處。諸法中實。不同虛空。性自神解。故名爲心。".

6   https://cbetaonline.dila.edu.tw/search/?q=%E6%80%A7%E8%87%AA%E7%A5%9E%E8%A7%A3&lang=zh (accessed on 6 July 2023).

7   『大乘起信論』 (T32, 578a12-13), "唯癡滅故。心相隨滅。非心智滅。".

8   『起信論疏』 (T44, 216c26-28), "非心智滅者。神解之性名爲心智。如上文云智性不壞。是明自相不滅義也。".

9   "自眞相者。......是依不異義門說也。", The unique true characteristic 自眞相 [of the *Lengqie abatuoluo baojing* 楞伽阿跋多羅寶經] is the unique characteristic 自相 of the *Rulengqiejing* 入楞伽經. The unique true characteristic is that the mind of original enlightenment mystically understands it by the nature itself, not a faulty indirect cause. This is based on the aspect of "not one" 不一. In addition, the unique true characteristic is that the nature of mystical understanding is not different from the original when [the mind] occurs "arising and ceasing" by the wind of nescience. This is based on the aspect of "not different" 不異.

10  The exact date when the *Shilun* was published is not clear, but it is likely that the *Shilun* was written earlier than the *Zongmishu* because Zongmi mentioned the title of *Shilun* in his writing, the *Yuanjuejing lueshu chao* (圓覺經略疏鈔, *Abridged Subcommentary to the Sutra of Perfect Enlightenment*) [X9.925c13].

11  The author is recorded as Nāgârjuna 龍樹, but the *Shilun* is regarded as an apocryphal scripture written in China or Korea around the end of the seventh century or the beginning of the eighth century. In addition, some Japanese books such as Shittanzō 悉曇藏 noted down the hearsay that the writer is the Silla monk Wŏlch'ung 月忠[T84.374c7].

12　『釋摩訶衍論』 (T32, 629c12-18), "謂一切諸眷屬染法。皆悉各各有二義故。云何爲二。一者神解義。二者暗鈍義。神解義者。據從本覺流轉邊故。暗鈍義者。據從無明流轉邊故。依初門故建立識名。依後門故建立相名。二門差別應如是知。何故如是。所言識者。解了義故順於本覺。所言相者，背本義故順於無明。" For more information on the *Shilun* and the commentaries on the *Shilun*, see (Morita 1935) and (Nasu 1992).

13　『釋摩訶衍論贊玄疏』 (X45, 836a12-14), "眷屬染法各具二義 一神解義始從本覺勢分發起立名為識識是了達順本覺 故 二闇鈍義始從 無明勢分發起立名為相相是背本順無明故"; (X45, 889c22-890a2), "云何(至)順於無明。釋曰次重微釋凡諸染法各具二義一者神解。神解勢力本覺所發所成之識似本覺故二者闇鈍勢力無明所發所成之相似無明故故分相識二甚別耳。".

14　『釋摩訶衍論贊玄疏』 (X45, 890a10-12), "釋曰 三染淨因緣望三細識 本覺親因無明疎緣 故所生果神解似覺 望三細相 無明親因本覺 疎緣 故所生法闇似無明。".

15　『釋摩訶衍論贊玄疏』 (X45, 890a9); 『釋摩訶衍論』 (T32, 629c26-630a1), "以何義故。根本無明隨染本覺各具因緣。互相望故。此義云何。謂舉本覺及與無明望於三識。本覺為因。無明為緣。同舉彼二望於三相。無明為因。本覺為緣。所以者何。以由親為因。由疎為緣故。".

16　『大乘起信論』 (T32, 576b7-9), "心生滅者。依如來藏故有生滅心。所謂不生不滅與生滅和合。非一非異。名為阿梨耶識。" The other one time is used in 『釋摩訶衍論記』 (X46, 79c8-12), "謂一下依義釋成二初通明相識二初解神暗一切眷屬染法皆依本覺無明二法力起識依本覺氣分性自明了故是神解義相依無明氣分性自漠冥故是暗鈍義由真妄力殊故神暗義別二。".

17　『釋摩訶衍論記』 (X46, 58c1-59a5), "五業相業識識相即暗鈍識即神解... 六轉相轉識識... 又有見名識謂神解故無見名相謂暗鈍故。七現相現識識... 又別異名識謂神解故相異名相謂闇鈍故... 十染淨始覺... 問二種本覺二種始覺及性真如皆說名識虛空無為何不爾耶答識者神解明了之稱虛空闇鈍無明了用是故不說。".

18　See (Kim 2015), p. 52 (Table 2); p. 54 (Table 4). I arbitrarily inserts underlines to indicate the same part.

19　『大方廣佛華嚴經隨疏演義鈔』 (T36, 235a20-23), "曉公釋云 本覺之心不藉妄緣，性自神解，名自真相，約不一義說。又隨無明風作生滅時，神解之性與本不異，亦名自真相，是依不異義說。".

20　『起信論疏筆削記』 (T44, 330a24-b2), "不同下約靈鑒以解心。謂虛空體亦無二邊。亦非差別虛相。然但昏鈍而無靈鑒。今此實性自在靈通。覺了不昧故云不同等... 斯則體相不二故。云一中實。神解故云心。".

21　『起信論疏筆削記』 (T44, 337b9-10), "心亦下三法合。神解者。本覺不昧。鑒照靈通也。餘文可知。".

22　『起信論疏筆削記』 (T44, 406a12-15), "涅槃真法入乃多塗。論其急要不過止觀。止乃伏結之初門。觀乃斷惑之正要。止乃養心識之善資。觀則照神解之妙術等。".

23　『大乘起信論疏略』 (X45, 444b19-20), "西京太原寺沙門法藏造疏　明南嶽沙門德清纂略。".

24　『楞嚴經正脉疏懸示』 (X12, 182b); 『大乘起信論義記』 (T44, 245a); 『大乘起信論疏』 (L141, 87b). 『十不二門指要鈔詳解』 (X56, 471b); 『大乘起信論疏』 (L141, 85b-86a).

## References

### *Primary Sources*

*Awakening of Faith in Mahāyāna* 大乘起信論, T32. no. 1666.
*Ba daren jue jing shu* 八大人覺經疏, X37. no. 673.
*Da banniepan jing jijie* 大般涅槃經集解, T32. no. 1763.
*Dafangguang fo huanyan jing suishu yanyi chao* 大方廣佛華嚴經隨疏演義鈔, T36. no. 1736.
*Dasheng qixinlun shulue* 大乘起信論疏略, X45. no. 765.
*Goryeoguk bojo seonsa susim gyeol* 高麗國普照禪師修心訣, T48. no. 2020.
*Guanzizai pusa ruyilun tuoluoni jing lueshu* 觀自在菩薩如意心陀羅尼經略疏, X23. no. 447.
*Haedong kosŭngjŏn* 海東高僧傳, T50. no. 2065.
*Huayanjing tanxuan ji* 華嚴經探玄記, T35. no. 1733.
*Huayan xuan tanhui xuanji* 華嚴懸談會玄記, X8. no. 236.
*Kisillon so* 起信論疏 of Wŏnhyo 元曉, T44. no. 1844.
*Lengqie abatuoluo baojing* 楞伽阿跋多羅寶經, T16. no. 670.
*Lengyan jing zhengmaishu xuanshi* 楞嚴經正脉疏懸示, X12. no. 274.
*Qixinlun jieyao* 起信論捷要, X45. no. 763.
*Qixinlun shu* 起信論疏 of Fazang 法藏 [大乘起信論義記], T44. no. 1846.
*Qixinlun shu* 起信論疏of Zongmi 宗密, L141. no. 1600.
*Qixinlun shu bixueji* 起信論疏筆削記, T44. no. 1848.
*Qixinlun shuji huiyue* 起信論疏記會閱, X45. no. 768.
*Qixinlun zuanzhu* 起信論纂註, X45. no. 762.
*Rulengqie jing* 入楞伽經, T16. no. 671.
*Shi buermen zhiyao chao xiangjie* 十不二門指要鈔詳解, X56. no. 931.
*Shiermenlun zongzhi yiji* 十二門論宗致義記, T42. no. 1826.
*Shi moheyan lun* 釋摩訶衍論, T32. no. 1668.
*Shi moheyan lun ji* 釋摩訶衍論記 of Puguan 普觀, X46. no. 774.
*Shi moheyan lun ji* 釋摩訶衍論記 of Shengfa 聖法, X45. no. 770.

*Shi moheyan lun shu* 釋摩訶衍論疏, X45. no. 771.
*Shi moheyan lun zanxuanshu* 釋摩訶衍論贊玄疏, X45. no. 772.
*Shittanzō* 悉曇藏, T84. no. 2702.
*Weimo jing lue shou* 維摩經略疏, T38. no. 1778.
*Xu gaoseng zhuan* 續高僧傳, T50. no. 2060.
*Yuanjuejing dashu* 圓覺經大疏, X9. no. 243.
*Yuanjuejing lueshu chao* 圓覺經略疏鈔, X9. no. 248.
*Zongjing lu* 宗鏡錄, T48. no. 2016.

### Secondary Sources

Hakeda, Yoshito S., trans. 1967. *The Awakening of Faith*. New York: Columbia University Press.
Kim, Cheonhak. 2015. Chongmire mich'in wŏnhyoŭi sasangjŏng yŏnghyang taesŭnggishillonsorŭl chungshimŭro 종밀에 미친 원효의 사상 [Wŏnhyo's Effect on Zongmi's Thought: Focusing on *Dashen Qixinlun Shu*]. *Pulgyohakpo* 불교학보 [*Journal of Institute for Buddhist Culture*] 99: 41–62.
Kim, Cheonhak. 2018. Pŏpchanggwa chongmil kishillonsoŭi yujŏn'gwa sasangjŏng sangwi 法藏과 宗密 「起信論疏」의 流傳과 思想的 相違 [A Study on the Circulation of Fazang's *Qishinlunshu* and its Ideological Differences from Zongmi's]. *Pojosasang* [*Journal of Bojo Jinul and Buddhism*] 51: 79–110.
Kim, Heejung. 2006. Wijindae shin'gaenyŏm yŏn'gu魏晉代 神概念 研究 [The Study on the Idea of Shen 神 in Wei and Jin Dynasty of China]. *Chungguksayŏn'gu*中國史研究 [*Journal of Chinese Historical Researches*] 41: 137–53.
Kim, Jiyun. 2021. Chunggugesŏ pŏpchang kishillonsoŭi yut'onge taehaesŏ 중국에서 법장 『기신론소』의 유통에 대해서 [The Study on the Distribution of Fazang's *Qixinlun shu* in China]. *Pulgyohakpo* 불교학보 [*Journal of Institute for Buddhist Culture*] 94: 1–28.
Ko, Youngseop. 2008. Wŏnhyo ilshimŭi shinhaesŏng punsŏk 원효 일심의 신해성 분석 [On Wŏnhyo's Concept of "Mystical Understanding of One Mind". *Pulhyohakyŏn'gu* 불교학연구 [*Journal for Buddhist Studies*] 20: 165–190.
Morita, Ryusen. 1935. *Shaku Makaen ron no kenkyū* 釋摩訶衍論之研究 [*The Study on the Shi Moheyan Lun*]. Kyoto: Fujii Sahee Press.
Nasu, Seiryu. 1992. *Shaku Makaen ron kogi* 釋摩訶衍論講義 [*The Note of Lecture about the Shi Moheyan Lun*]. Narita: Naritasan Bukkyo Kenkyujo.

