# Peer review of "The Influence of Wŏnhyo’s Understanding of “Shenjie” 神解 on the Chinese Commentaries on the Awakening of Faith in Mahāyāna"

_religions, doi:10.3390/rel14070904_

Round 1

Reviewer 1 Report

I thank the author for a very interesting read. This paper will be a significant contribution to what remains an understudied field. I found the conclusion to be particularly strong.

What currently holds this paper back is the overall cohesiveness of the writing. The paper requires a thorough proofread for typos and missing words (eg. "Faang" on line 153, missing "the" on line 252 etc.) and I would recommend that the author ensures that the argument that they want to make is as clear as it can be. Regarding the latter, the paper in its current state is oftentimes clunky to the extent that the argument is unclear: lines 348-351, for example, is unclear.

With proofreading and re-phrasing, this will constitute an excellent paper. My thanks to the author for a fascinating read!

Reviewer 2 Report

The Romanization of Korean should be consistent.

It seems that the author wants to employs the McCune-Reischauer method. But lots of errors are found there.

In the abstract:

Line 9: influence “on Fazang” instead of “of Fazang”

Line 10 Wonhyoso - >Wŏnhyoso et passim. Make the necessary changes wherever this happens.

Line 10: The sentence “First, comparing the usage of shenjie in the Fazangshu and the Wonhyoso identified the characteristic interpretation of the latter.” leaves a lot to be desired. Please rewrite this sentence to make the meaning more precise.

1: 34 “the Silla monk” instead of “Silla Monk”

1: 41 “supernatural and godly”? (to make it balanced?) Cf. It is also sometimes translated as “numinous”

2: 48 : It should be “T'aehyŏn”

2: 54 add “the’ after “Wŏnhyo on”

2: 78: “Fazang follows the Wonhyoso in many parts but presents his interpretation when his opinion differs from Wŏnhyo, like with the seventh consciousness and manas, and the usage of shenjie is one of them.” This sentence can be improved. How about the following? “Fazang follows the Wŏnhyoso in many parts but presents his own alternative interpretation when his opinion differs from Wŏnhyo, e.g., when he analyzed the concept of the seventh consciousness and manas, and the usage of “shenjie” is one of them.” I put “shenjie” in the quotation mark because it is the term that is mentioned and discussed, not used to refer to the state of the world/mind. As a matter of fact, I think all occurrences of “shenjie” in the body of the draft should be like this or perhaps italicized. Also as for “manas” do you mean the seventh consciousness, i.e., kliṣṭa-manas, in other words, kliṣṭamanovijñāna? Or just one of the terms referring to the mind together with the others being “citta” and “viññāṇa”?

2:88 It should be: “Haedong kosŭngjŏn”

3. 94, 102, 105 et passim, “two approaches” vs. “two aspects.” It seems that they are both translations of one and the same word “”. If this is the case, this double translation would be a big mistake, potentially confusing the readers without any knowledge of the Chinese character.

3: 112 “Cheonhak, 2018” should be corrected.

4: 151 “nor in the state..” instead of “nor the state..”

2.3 Title: “Fazang” instead of “Faang”

4 Table 1. Middle “whole” instead of “shole”?

5: 185 “analyzes”? instead of “defines”

6: 236 “suggest” instead of “representation”

7: 239-241. This sentence is not well written. Rewrite this.

7: 268-269 This sentence is not well written. Rewrite this.

7: 274-275 This sentence is not well written. Rewrite this.

7: 287-292 “tells” “says” etc are not the best translations here.

8: 338 “Cheonhak, 2015” should be corrected.

12: 441 “Jiyun 2021” - -> “Kim, Chiyun 2021”?

12: 454 Wonhoso - - > Wŏnhyoso

12: 469 “keyword” - - > “one of the keywords”

16 Secondary sources

The last names and the first names are inverted for the Korean names.

Put the last name first and the first name. Cite the last name and if necessary provide first name in the body.

Round 2

Reviewer 2 Report

Please read the comments attached below

Author Response

Response to Reviewer Comments

Thank you very much for your very kind review.

 Point 1 : General Comments

I changed the Korean titles, such as ‘McCune-Reischauer Romanized form / Korean / English translation. In addition, the Japanese titles also changed.

Point 2: Specific Comments

I changed Line 11 & Line 453(corresponding to revised version line 600) as you recommended.

Point 3: the Secondary Literature Part

I put a comma after the last name and changed the order of the name of Hakeda.

Point 3: *****

I tried to find the answer to your question, but I could not because this is a very complicated issue. I assume that Fazang’s usage of “Shenjie” is related to the relationship between Huayan 華嚴 and other orders 宗 in China. Since this requires examining various matters, such as Fazang’s thoughts and the circumstance of the Buddhist society in the era when Fazang lived, the 7th -8th century. Therefore I try to find the answer to this question in the next study.

Round 3

Reviewer 2 Report

After two substanatial revisions, I now recommend its publication.